# Implications of Diet and The Gut Microbiome in Neuroinflammatory and Neurodegenerative Diseases

**DOI:** 10.3390/ijms20123109

**Published:** 2019-06-25

**Authors:** Sarah Hirschberg, Barbara Gisevius, Alexander Duscha, Aiden Haghikia

**Affiliations:** Department of Neurology, Ruhr-University Bochum, St. Josef-Hospital Bochum, Gudrunstr. 56, 44791 Bochum, Germany; sarah.hirschberg@rub.de (S.H.); barbara.gisevius@rub.de (B.G.); alexander.duscha@rub.de (A.D.)

**Keywords:** multiple sclerosis, microbiome, neurodegeneration, gut–brain-axis, short chain fatty acids, dysbiosis, neuroinflammation

## Abstract

Within the last century, human lifestyle and dietary behaviors have changed dramatically. These changes, especially concerning hygiene, have led to a marked decrease in some diseases, i.e., infectious diseases. However, other diseases that can be attributed to the so-called ‘Western’ lifestyle have increased, i.e., metabolic and cardiovascular disorders. More recently, multifactorial disorders, such as autoimmune and neurodegenerative diseases, have been associated with changes in diet and the gut microbiome. In particular, short chain fatty acid (SCFA)-producing bacteria are of high interest. SCFAs are the main metabolites produced by bacteria and are often reduced in a dysbiotic state, causing an inflammatory environment. Based on advanced technologies, high-resolution investigations of the abundance and composition of the commensal microbiome are now possible. These techniques enable the assessment of the relationship between the gut microbiome, its metabolome and gut-associated immune and neuronal cells. While a growing number of studies have shown the indirect impact of gut metabolites, mediated by alterations of immune-mediated mechanisms, the direct influence of these compounds on cells of the central nervous system needs to be further elucidated. For instance, the SCFA propionic acid (PA) increases the amount of intestine-derived regulatory T cells, which furthermore can positively affect the central nervous system (CNS), e.g., by increasing remyelination. However, the question of if and how PA can directly interact with CNS-resident cells is a matter of debate. In this review, we discuss the impact of an altered microbiome composition in relation to various diseases and discuss how the commensal microbiome is shaped, starting from the beginning of human life.

## 1. Introduction

In recent years, the implications of the gut microbiome and its metabolites in the development and progression of various diseases has gained increasing attention. Therefore, in this review we aim to give an insight into the current research regarding microbiome alterations as a possible cause for the increase in autoimmune and neurodegenerative diseases. Environmental factors are frequently discussed as triggers or even causes of various diseases. Although understanding the direct connection between environmental factors and disease incidence is still a matter of debate, nutrition and the continuously varying dietary habits in Western society are already established as a causative link to disease accumulation.

In humans, the microbiome of the gut, particularly of the colon, harbors the largest number of microorganisms [1]. Within the colon, among other functions, the bacterial population processes indigestible components of food by bacterial-derived enzymes, optimizing the uptake of nutritive sources [2,3]. During this process, gut bacteria produce a vast amount of bioactive compounds that are crucial for health and disease in the host [4]. This symbiosis comprises pivotal developmental advantages in eukaryote evolution, not only based on the improved fermentation of food, but also on the direct interaction between the microbiome and the innate and adoptive immune system. Therefore, a so-called dysbiosis, i.e., an imbalance in the microbiome composition, exerts deleterious consequences on human health and immunity [5]. A dysbiotic microbiome composition can be the result of several external and internal factors, e.g., treatment with antibiotics, stress, or diet [6,7,8]. However, not only external factors, but also the host’s genetics have an impact on the gut microbiome composition [9].

Various chronic disorders are considered to be associated with an altered microbiome. In inflammatory bowel disease (IBD), a compositional shift in the commensal microbiome is caused by a chronic inflammation of the gastrointestinal tract [10,11]. In addition, the expansion of potentially harmful bacteria, the so-called pathobionts, is discussed as an underlying cause of IBD, namely in Crohn’s disease [12]. Besides the main bacterial phyla, i.e., *Firmicutes*, *Bacteroidetes*, and *Actinobacteria* [13], pathobionts in low abundance also normally occur in a healthy individual, but only tend to expand under pathological conditions, as is the case in the dysbiotic state [12,14]. The chronic auto-immune disease celiac disease (CD) is triggered by the consumption of gluten and affects the gastrointestinal system. Recently, the impact of the gut microbiome as a modulator of the disease-associated immunopathology was discussed [15]. CD patients are mostly positive for HLA-DQ2 or HLA-DQ8 [16], but not every person with this susceptibility necessarily develops the disease [15]. Therefore, the investigation of additional disease-associated factors is on the rise. Various alterations of the gut microbiota in CD have been found, and so far it has been demonstrated that patients with active CD show a decrease in the abundance of *Firmicutes* [17], as well as a decrease in the proportion of *Bifidobacterium* [18,19,20,21], *Lactobacillus spp*. [20], and *Enterococcus* [20]; instead, patients have an increase in the abundance of *Proteobacteria* [17,22], and increased levels of *Bacterioides* [19] [20,23], *E. coli* [19,23], *Staphylococcus* [19,20], and *Clostridium* [19]. The symptomatology of CD does not only comprise gastrointestinal problems, but also neurological impairments. The most common neurological manifestation of CD is cerebellar ataxia, followed by peripheral neuropathy, as well as many other neurological symptoms, e.g., epilepsy and cognitive impairment [24]. Results from electroencephalography (EEG), somatosensory evoked potentials (SEPs), and transcranial magnetic stimulation (TMS)—which show epileptic discharges, dysfunctional somatosensory conduction, disinhibition and hyperfacilitation of the motor cortex—lead to the introduction of a so-called “hyperexcitable celiac brain” [24].

The commensal microbiome is essential for the regulation of immune tolerance [25]. Regulatory immune components, such as regulatory T cells (Treg), ensure the sufficient establishment of immune tolerance, not only towards endogenous components, but also more specifically towards the enteric microbiota [26]. The immune system directly influences the commensal microbiome and vice versa. Therefore, it is not surprising that autoimmune diseases have been shown to be associated with a dysbalanced microbiome. For instance, recent studies discovered alterations in the microbiome of patients suffering from the autoimmune disease multiple sclerosis (MS) [27]. In MS, distinct bacterial phyla, which are supposed to be involved in the induction of intestinal Treg, were decreased in comparison to healthy controls [28].

Also, in psychiatric disorders, including autism spectrum disorder (ASD) and depression, associations with gut microbiome alterations were found [29,30]. In ASD, gastrointestinal disruption correlates with disease severity [31]. Additionally, a reduction in microbiome diversity, characterized by reduced abundances of the genera *Prevotella*, *Coprococcus*, and unclassified *Veillonellaceae*, was observed [29]. ASD-dependent behavioral as well as gastrointestinal symptoms could be ameliorated by fecal microbiota transplantation [32]. However, interventions with fecal transplantations or probiotics are still a matter of debate, due to a lack of temporal and causal relationship between microbial alterations [33]. It is not known whether these interventions actually have the ability to reverse already established metabolic or immunological deficits. Additionally, it requires to be further investigated which microbial composition specifically cures or even worsens a certain disease [33].

Furthermore, a dysbiotic state is also discussed in relation to primary and secondary neurodegenerative diseases. Most neurodegenerative disorders have been associated with a genetic predisposition. Although the involvement of environmental factors in disease pathogenesis has been investigated for decades, currently, the impact of diet and its consequences on the microbiome have only recently been appreciated.

## 2. The Role of Diet in Dysbiosis Onset

The symbiotic relationship between eukaryotes and prokaryotes started early in evolution. One of the earliest and most impressive examples are the mitochondria, which originate from endosymbiotic prokaryotes [34,35,36]. Furthermore, to date, the human body is in a symbiotic relationship with trillions of bacteria, which have remained autonomously. These bacteria colonize the skin and the mucosal sites, i.e., the pulmonary system, and the gastrointestinal tract where they ensure the digestion of complex food components, protection from pathogens, and the establishment of a healthy immune system [37]. In humans, colonization with bacteria is assumed to occur during or shortly after birth. However, currently a possible pre-birth colonization has also been discussed, supported by the observation that bacterial DNA is present in the placenta [38,39,40]. Already during birth, delivery via caesarean section (C-section) or via natural birth significantly determines the abundance and amount of gut bacteria [41,42]. In addition, whether newborns are breast-fed or nursed with formula milk contributes to their microbial composition later on [43,44]. The microbiome composition evolves within the first three years of life and remains until adulthood [44,45]. In addition, the implication of the microbiome in neurodevelopmental processes has already been shown. In a mouse model, a maternal high-fat diet caused alterations in the offspring’s social behavior, based on a reduction of oxytocin-immunoreactive neurons, along with reduced long-term potentiation of dopaminergic neurons within the ventral tegmental area [46]. On the contrary, a diet containing omega-3 polyunsaturated fatty acids beneficially influences stress resistance and depressive-like behavior mediated by the hypothalamic–pituitary–adrenal (HPA) axis [47].

The diversity of the gut microbiome can be influenced by several factors. It has been shown that the phylogenetic composition of fecal microbiota differs between human populations of different countries, under non-pathological conditions. A comparison of United States (US) residents with Malawian and Amerindian cohorts showed that in US adults the fecal microbiota was observed to be the least diverse. This variance, beside differences in geography, can be due to diet, hygiene and lifestyle in general [45]. A factor of increasing interest is diet. For instance, long-lasting dietary alterations even alter enterotype states. Investigations of microbiome-associated nutrients displayed distinct correlations between certain diets and the occurrence of bacterial phyla. Predominantly, the phyla *Bacteroidetes* and *Actinobacteria* are positively associated with fat, but negatively associated with fiber; however, the opposite association was discovered for *Firmicutes* and *Proteobacteria* [48]. Since short chain fatty acids (SCFAs) are the major metabolites produced by bacteria in the gut, a dysbiosis is mostly associated with a decreased abundance of SCFA-producing bacteria. Reduced amounts of SCFAs are associated with a rather inflammatory environment and a disrupted gut epithelial integrity [49]. In a viscous circle, alterations in diet cause imbalances in bacterial communities, which in turn change metabolic processes within the microbiome, thereby causing a bias in metabolites, which are made available for the human organism. These metabolites include not just SCFAs but also bile acids, lipids and vitamins. SCFAs, for instance, manipulate the host´s gene expression by acetylation and methylation, thereby intervening in cellular and metabolic processes [50]. Most interestingly, diet directly influences histone acetylation and methylation of multiple host tissues, whereby low abundances of SCFAs are associated with a lesser degree of acetylation, representing changes in chromatin remodeling [51]. 

Our main food components consist of carbohydrates, proteins, and fats. The main end products after bacterial fermentation are SCFAs [52], micronutrients such as vitamins [53], and secondary bile acids [54]. Organic acids, emerging from the fermentation of carbohydrates, are used as an energy source by the resident bacteria. After passing through the gastrointestinal tract endothelia, either by active transport or passing passively, the metabolites enter the blood circulation [4]. Once systemically available, these metabolites have a crucial impact on various physiological processes. For instance, the consumption of high amounts of dietary fibers has been shown to reduce the risk for coronary heart disease [55] and type II diabetes [56], for example, by lowering total cholesterol and low-density lipoprotein cholesterol [57] and by reducing fasting blood glucose and glycosylated hemoglobin [58], respectively. SCFAs exert various functions, mainly by directly regulating the immune response [4]. Besides diverse dietary interventions, in other studies, SCFAs have been shown to directly regulate the immune response, thereby rather favoring an anti-inflammatory environment. For instance, butyrate and propionate have the ability to induce apoptosis of neutrophils [59]. Additionally, it was shown that SCFAs increase the amount of colonic Treg [60]. In an animal model of colitis, microbe-derived butyrate was found to be relevant for the induction of the extrathymic generation of Treg [61], and ameliorated the disease course by inducing the differentiation of colonic Treg [62]. Butyrate also exerts immunomodulatory effects on intestinal macrophages, causing a reduction in the production of pro-inflammatory cytokines, which is mediated by histone deacetylase (HDAC) inhibition [63].

Moreover, SCFAs are important drivers of metabolic processes. In a fecal microbiota transplantation study, the infusion of intestinal microbiota from lean donors to patients with metabolic syndrome resulted in an increased insulin sensitivity coupled with elevated levels of butyrate-producing bacteria [64]. Additionally, both propionate and butyrate have been shown to positively modulate body weight and glucose homeostasis by activating intestinal gluconeogenesis [65].

Since an un-physiologic diet is one of the main issues in large areas of the world today, the term Western diet has evolved into a paradigm for this kind of diet, based on the diet-dependent accumulation of chronic disorders and obesity [66,67]. The Western diet is characterized by high amounts of animal proteins, simple sugars and fats, and mostly lacks the consumption of fibers [4]. 

The link between diet, the gut microbiome, and the incidence of chronic multifactorial diseases, such as neurodegenerative disorders, represents a more complex interplay. In an experimental model of MS, for instance, we could show that the administration of the SCFA propionic acid (PA), which is produced by gut bacteria during the fermentation of fiber-rich food compounds, ameliorates the disease course. Hence, reduced inflammatory processes within the central nervous system (CNS) are accompanied by less demyelination of the white matter and less axonal loss. In contrast, the administration of long chain fatty acids (LCFAs), widely found in the diet, exacerbates disease symptoms. Most interestingly, based on the strong relationship between gut bacteria and the cells of the gut-associated lymphatic tissue, PA increases the amount of intestine-derived anti-inflammatory Treg, whereas LCFAs promote the polarization towards pro-inflammatory T-helper (Th) 1 and Th17 cells [60]. Further, regarding the direct interplay between microbial peptides and the gut-associated immune system, functional observations could show that specific microbial protein-T cell-Receptor (TCR) interactions lead to the induction of a MS-like disease, including axonal loss, mediated by epitope mimicry in a humanized mouse model [68]. Besides fatty acids, other studies have also shown that high salt diet—another crucial component of the Western diet—exacerbates disease activity of experimental autoimmune encephalomyelitis (EAE) by Th17 induction via the depletion of Lactobacillus murinus [69]. 

The use of probiotics to overcome a dysbalanced microbiome may pose a therapeutic option. Probiotics are “live organisms that, when administered in adequate amounts, confer a health benefit on the host” [70]. Thus, besides the influence on probiotic species by dietary changes, the gut microbiota composition could be directly modulated by certain species or a mixture of species to evoke health benefits. So far, reported benefits include protection against infectious disease, shown by reduced sepsis in infants in rural India in a randomized synbiotic trial [71], the prevention of cardio-metabolic disease, represented by a decrease in glucose, HbA1c, insulin and homoeostasis model assessment-estimated insulin resistance (HOMA-IR) in patients with diabetes [72], and the alleviation of gastrointestinal symptoms, reported as an increase in gastrointestinal well-being and decrease in digestive symptoms [73]. By now, the only approved fecal microbiota transplantation therapy exists for *Clostridium difficile* infection and could act as an alternative to antibiotic treatment in primary *C. difficile* infection [74]. Probiotics as dietary supplements also show a high degree of controversial findings. Whereas no positive effects of probiotic use have been reported in Crohn’s disease, in ulcerative colitis, probiotics are effective as remission induction and maintenance therapy [75]. Probiotic treatment efficacy is highly context dependent, for instance, it can differentially affect people with the same health status. In a recent article, it was reported that the gut mucosal colonization of humans, who consumed a mixture of eleven probiotics, differed inter-personally. The probiotics were found in the feces, but regarding the mucosal microbiome, there was a person-, strain- and region-specific colonization resistance, which led to the probiotic’s differentiated ability to affect the resident mucosal microbiome and its function [76]. This study highlights possible limitations that can occur when only microbiome analysis from fecal samples are taken into account, instead of combining it with functional analysis of the mucosal microbiome. Therapeutical interventions addressing the microbiome by targeting dietary habits seem to be a promising tool to expedite research concerning so-called precision medicine, comprising diagnostics and treatment. However, standardized protocols for the collection and isolation of the microbiome, as well as sequencing techniques, are required in order to ensure a successful applicability [77]. 

## 3. Dysbiosis in Neurodegenerative Disorders

The impact of microbiome dysbiosis in the development of neurodegenerative disorders has gained increasing attention of the current research. The main challenge here is to breakdown the causal relationship of cause and consequence—which is by now rarely discussed, due to the intricacy of the human body and heterogeneity of chronic disorders. First, associations between an altered microbiome composition and neurodegeneration were provided for primary and secondary neurodegenerative disorders. For instance, in Parkinson´s disease (PD), the abundance of the genera *Prevotellaceae*—known as SCFA-producers—was shown to be reduced [78]. Furthermore, PD patients show reduced levels of the SCFAs acetate, butyrate, and propionate in fecal samples, in comparison to age-matched controls [79]. Also, the relative abundance of *Enterobacteriaceae* could be associated with the severity of motor dysfunction in PD [80]. More and more studies are describing gastrointestinal microbiome alterations in PD patients compared to non-PD patients, but the bacterial species altered show a high variety between these studies [81], possibly based on differences in study population, methodological implementation, and research focus. In Alzheimer´s disease (AD), 16S ribosomal RNA gene sequencing of DNA isolated from fecal samples revealed a decreased abundance of *Firmicutes* and *Actinobacteria* and an increased abundance of *Bacteriodetes* at the phylum level. Furthermore, a correlation between the altered abundance and biomarkers in the cerebrospinal fluid of AD pathology was observed [82]. Additionally, the Gram-negative bacterium *Helicobacter pylori* is positively associated with, and seems to influence the course of, AD [83,84]. It is hypothesized that *H. pylori* induces blood–brain barrier (BBB) dysfunction in AD by increasing the blood homocysteine level, followed by hydrogen peroxide production due to homocysteine auto-oxidation. Hydrogen peroxide subsequently damages vascular endothelial cells of the BBB, leading to its dysfunction. As a consequence, the Aβ concentration in the brain increases with a concurrent decrease in its clearance [84]. 

Microbiome metabolites can modulate the disease severity of neurodegenerative diseases via two mechanisms: (a) by immune-mediated neurodegeneration, or, (b) by direct effects of microbiome-derived metabolites on cells of the CNS (Overview of gut-brain interaction in Figure 1). Concerning indirect effects, it has been shown that Treg are able to exert neuroprotection by increasing remyelination and oligodendrocyte differentiation [85]. Moreover, interleukin (IL) 10, as a key Treg cytokine, is capable to trigger neuroregeneration [86]. In contrast, in EAE it was shown that pro-inflammatory Th17 cells directly promote neurodegeneration mediated by the direct interaction between Th17 and neuronal cells, especially during the peak of disease. This interaction causes severe neuronal damage [87]. The direct effects of microbiome-derived metabolites on the CNS are currently poorly understood, but no less important. Since it is known that gut microbes produce neurotransmitters like γ-Aminobutyric acid (GABA) [88], histamine [89], dopamine, noradrenaline and serotonin [90,91], there is probably a large number of other neuro-active agents produced by gut bacteria that can directly interact with cells of the CNS. In detail, for the inhibitory neurotransmitter GABA, it was shown that GABA can be produced by strains of *Lactobacillus* and *Bifidobacterium* through the decarboxylation of l-glutamate by glutamate decarboxylases [88]. This synthesis, from l-glutamate to GABA, also takes place in inhibitory neurons [92]. Only a few studies have so far discussed the effect of gut microbiome-produced metabolites and their interaction with the CNS. For example, a chronic treatment with *Lactobacillus rhamnosus* induced changes in CNS GABA receptor expression in mice. The modulation of GABA_Aα2_, GABA_Aα1_ and GABA_B1b_ mRNA expression, as well as a reduction in anxiety- and depression-related behavior by *Lactobacillus rhamnosus*, could be reversed by vagotomy [93]. GABA can be transported from the gut lumen into the blood through the hPAT1 H+/amino acid symporter, which is present in the human gastrointestinal tract [94]. However, the existence of GABA transporters in the human BBB, and hence the ability of gut-derived GABA to enter the CNS, is still under evaluation. Since the variety of resident bacterial strains seems to be endless, so are the microbiome-derived metabolites. The study of these potential neuro-active compounds is still in its infancy, especially the use of humanized in vitro models, which is more of an exception than a rule. 

The interplay of gut metabolites and the CNS has shaped the term of the gut–brain axis. This interplay was classically understood to be one-directional, meaning that the CNS exclusively regulates gastrointestinal functions. The modulation of the gut function, namely the motility or the secretion of components into the gut lumen, is regulated by the sympathetic and the parasympathetic nervous system [95]. Besides the sympathetic and parasympathetic nervous system, the vagus nerve together with the enteric nervous system (ENS) and the hypothalamic–pituitary–adrenal axis (HPA), originally shaped the term gut–brain axis. These components link the CNS with the visceral organs, including the gastrointestinal tract [96], referring to the bidirectional communication between the brain and the gastrointestinal system. The vagus nerve contains 20% efferent fibers, sending signals from the brain to the body, and 80% afferent fibers, sending signals from the periphery to the brain [97]. Thereby, different branches of efferent fibers orchestrate the intestinal contraction of smooth muscles and mucosal secretion within the gut [96] by providing inhibitory and excitatory stimuli [98]. Since afferent fibers of the vagus nerve are not directly connected to the gut microbiota, the interaction is limited to cells located in the epithelium or, concerning microbiome-derived signals, to metabolites that diffuse through the epithelium [99]. Enteroendocrine cells are part of the intestinal epithelial cells, modulating the motility, secretion and intake of nutrition within the gastrointestinal tract. By the release of several substances, e.g., serotonin, ghrelin, cholecystokinin, and peptide YY, enteroendocrine cells communicate with vagal afferent fibers, which express a variety of the relative receptors [100]. The communication of the ENS with cells of the intestine is mediated by a complex neural network [101], and thereby the gut functionality is modulated by synaptic transmission. Mechano- and chemoreceptors respond to various extrinsic stimuli. The motor activity of the stomach and the small intestine are modulated by serotonin-sensitive vagal mechanoreceptors, present within the afferent fibers [102]. 

Only recently has the underlying reciprocal impact of microbiome metabolites on the gut–brain axis gained increasing attention [103]. In this context, the term gut–brain axis has been extended to “microbiota–gut–brain axis”. Microbial metabolites are known to directly impact a subset of enteroendocrine cells in the gut [104]. For instance, gut-derived SCFAs are assumed to exert a variety of effects on the gut–brain axis. It is already known that SCFAs can affect several cell types [105,106]. These effects are either mediated by the G-protein coupled receptors (GPR) 41 and 43 [107], or by epigenetic modulation via HDAC inhibition [108,109,110]. For enteroendocrine cells, it was shown that these effects were mediated by GPR 41, thereby contributing to the energy balance of the host. More specifically, in GPR 41 knockout mice, the gut motility is increased due to reduced levels of peptide YY, resulting in a deficiency in normal weight gain [104]. The link between SCFAs and histone acetylation needs further investigation concerning the causal connection between environmental factors and response in gene expression. In graft-versus-host disease, microbiome alterations resulted in a reduced availability of the SCFA butyrate. A reduction of butyrate was accompanied by a decreased level of acetylated histones, whereas intragastric administration of butyrate restored histone H4 acetylation levels [111]. The acetylation of histones is considered to facilitate gene expression by alterations of the chromatin status. Therefore, HDAC inhibitors prevent the removal of acetyl groups from lysine residues of the chromatin structure by HDACs, therefore ensuring a relaxation of the nuclear chromatin structure. This relaxation promotes the interaction of transcription factors with gene promoters, which modulates gene expression [112]. There are four classes of HDAC inhibitors: class I inhibitors include the HDACs 1, 2, 3 and 8; and class II inhibitors, subdivided into class IIa including HDACs 4, 5, 7, 9, and class IIb including HDACs 6 and 10. Class IV HDACs include HDAC 11. Class III HDACs are differentiated based on their dependence on nicotiamide adenine dinucleotide (NAD+) and are commonly known as sirtuins, however, class I, II and IV HDACs are zinc finger-dependent enzymes [112,113]. In addition, acetylation is not exclusively limited to chromatin structures, but also occurs in various proteins, thereby intervening in various cellular processes, including protein expression and function, mitochondrial function, intracellular transport, and metabolic processes [114]. Post-translational modifications could play a key role in closing the gap between environmental factors and genetic disease susceptibility, especially, since imbalances between histone acetyltransferases (HAT) and HDAC activity are considered to favor neurodegenerative diseases [112]. On the one hand, it was already demonstrated that an altered HDAC activity causes neuronal damage. A member of the HDAC Classes I and II, namely HDAC1, induces axonal damage leading to neuronal death, following nuclear export [115]. On the other, alterations in the histone-acetylation status have been observed in neurodegenerative conditions. In Huntington´s disease (HD), the polyglutamine-containing domain of the abnormal huntingtin protein was able to bind to HAT domains, therefore influencing the acetyltransferase activity [116]. Further, investigating the expression of HDACs in the brains of HD patients, the loss of acetylated histone H2A, H2B, H3 and H4 within cells in the caudate nucleus and Purkinje cells of the cerebellum could be observed, while HDAC 5 levels were increased [117]. Additionally, in an animal model of HD, butyrate increased histone acetylation, thereby protecting against neurotoxicity. In contrast, acetylated histone H3 and H4 levels were significantly increased in post-mortem brain tissues of AD patients [118]. Correspondingly, in PD patients, alterations in histone acetylation could be also observed in post-mortem tissues [119]. Therefore, targeting histone acetylation processes could comprise a promising tool for novel therapeutic strategies.

In Alzheimer´s disease, targeting chromatin remodeling by using the HDAC inhibitor sodium 4-phenylbutyrate also exerts beneficial effects [120]. Likewise, sodium butyrate displayed neuroprotective effects in Huntington´s disease by deacetylase inhibition [121]. In addition, a recent study has shown that propionate protects against haloperidol-induced neurite lesions, possibly by modulating the transcription factor cyclic adenosine monophosphate (cAMP) response element-binding protein (CREB), thus preventing the reduced expression of neuropeptide Y [122]. PA beneficially affects the blood–brain barrier integrity by protection from oxidative stress via the nuclear factor, erythroid 2 like 2 (Nrf2) signaling pathway [123]. However, there is still a lack of explicit proof for the direct contribution of gut metabolites to neurodegeneration and neuroregeneration.

In summary, the composition and diversity of our gut microbiome is highly determined by our everyday life: the geographical sphere we life in, the infectious agents we have been in contact with, the antibiotics we use, and, probably most important of all, our diet. The interplay of SCFAs with the gut microbiome and the associated immune system is a highly interesting, but also intensely studied, topic of current research. Therefore, it is necessary to expand future investigations to the assessment of the currently unknown metabolites and their potential to directly interact, beyond the metabolome–immune axis, with cells of the CNS. The suspicion of a connection between various disorders, like neurodegenerative diseases, and the gastrointestinal milieu, is increasingly being confirmed. However, the question of to what extent the microbiome contributes to neurodegeneration is still not fully understood. Targeting the microbiome, either by dietary interventions or the administration of prebiotics, opens up new therapeutic avenues for the treatment of both systemic diseases and also neurodegenerative disorders. Even though the immunomodulatory capacity of gut metabolites provides an explanation for the amelioration of neurodegenerative processes, the direct interplay of gut metabolites on neurons or glia cells needs to be further investigated.

## Figures and Tables

**Figure 1 ijms-20-03109-f001:**
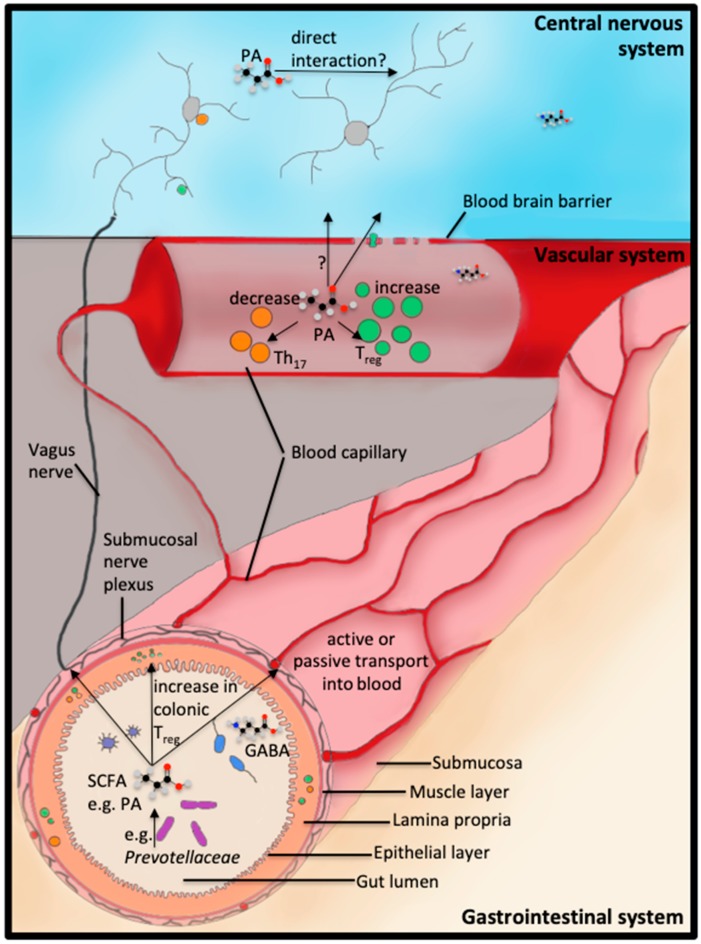
The gut and the brain—interacting systems. The central nervous system is in direct contact with the gastrointestinal system through the enteric nervous system and the vagus nerve. Furthermore, gut microbiome-produced metabolites, like the short chain fatty acid (SCFA) propionic acid (PA) or the neurotransmitter γ-Aminobutyric acid (GABA), can take the indirect route of the vascular system to finally reach the central nervous system (CNS).

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
