# Peer review of "Implications of Diet and The Gut Microbiome in Neuroinflammatory and Neurodegenerative Diseases"

_ijms, 2019, doi:10.3390/ijms20123109_

Reviewer 1 Report

Hirschberg et al. reported a review article regarding the implication of diet and the gut microbiome in relation to neuroinflammation and neurodegenerative disease. This field is very hot at present, and therefore, this review article is interesting. As authors mentioned, the mechanism by which gut microbiome affect the neurological function has much attenion but remained quite unclear. On the other hand, the diet is also an important theme because it largely affect the composition of gut microbiome but also metabolism. Although these two important theme is difficult to review satisfactorily, authors have well organized each section to explain those themes.

Author Response

We thank Reviewer 1 for reviewing our manuscript. 

Reviewer 2 Report

The authors deal with an intriguing timely topic, which is the diet-related impact on the gut microbiome and its effects on inflammatory diseases and disorders of the gut-brain axis. The findings here reviewed may have significant translational implications in clinical practice and future scientific research within this cutting-edge topic. Overall, the review is nicely conceived and designed; the results retrieved from the literature seem to be consistent and are adequately commented. However, there are some points needing attention and revision by the authors.

- The aim and rationale of this review should be clearly stated in the main text too.

- A table summarizing the main data from these studies would be helpful.

- The involvement of both gut microbiome (Verdu EF, et al. Nat Rev Gastroenterol Hepatol 2015) and gut-brain axis in celiac disease (Pennisi M, et al. Front Neurosci 2017) needs to be mentioned. In particular, clinical neurophysiology can contribute to the assessment and monitoring of celiac patients, even in those without a clear neurological involvement (Pennisi G, et al. PLoS One 2014). The evidences seem to converge on a “hyperexcitable celiac brain”, which partially reverts back only after a long-term gluten restriction (Bella R, et al. PLoS One 2015; Pennisi M, et al. PLoS One 2017). Therefore, given its potential neuroprotective effect, the gluten-free diet should be introduced as early as possible, although the overall response of neurological symptoms (and cognition in particular) is still controversial, as recently reviewed (Lanza G, et al. Int J Mol Sci 2018).  

- Limitations of the studies currently available and future research directions should be highlighted.

Author Response

The authors thank Reviewer 2 for the constructive suggestions, which will strengthen the aim of our review appropriately. Therefore, we included a sentence about the aim of the review into our introduction, which hopefully meets the reviewer’s expectation and helps to clearly state our concern.   

We agree, that a summarizing table in general is helpful to outline main findings. Due to the heterogeneity of the mentioned diseases and the fact that we talk about neurological right up to autoimmune disorders, we decided not to include a table, since we think this would be too superficial and would not deliver our message. Instead, we compiled a figure that demonstrates the relevant connections between the gut and its microbiome, the vascular system, and the central nervous system - which are relevant for all diseases mentioned in this review. We hope this rationale is sufficient for the reviewer.

We thank the Reviewer 2 for calling our attention to celiac disease (CD). The authors tried to shortly summarize the main aspects of gut microbiome alterations in CD and its neurological manifestation. We hope this is to the reviewer's satisfaction. 

Addressing limitations of the studies currently available, starting in line 98 up to line 100, we tried to state difficulties in implementing translational approaches, namely fecal transplantations. We further expanded this passage to outline these limitations more clearly. Additionally, starting in line 197, we tried to emphasize ongoing therapeutical approaches and their pitfalls. However, since our message seems to come off misleading, we added a conclusive sentence to hopefully clarify the aim of this passage. In the following, within the paragraph of dysbiosis in neurodegenerative diseases, occurring limitations in individual study designs were mentioned within the elaboration of the paragraph about Parkinson´s disease, hopefully underlining outstanding needs for future research and meeting the reviewers expectations.   

Reviewer 3 Report

In the present review article Hirschberg et al discussed about the role of microbiota alterations in the development of inflammatory and degenerative brain disorders. Main comments:

1) The content of the Abstract is too vague: please summarize few results reported in the main body of the paper.

2) The paragraph 2 analyzed the role of diet in dysbiosis onset, therefore I suggest to change the title of the paragraph.

3) Data about Parkinson disease are lacking.

4) Helicobacter pylori may be considered as a pathological component of human flora. There are some papers hypothesizing a relationship with Alzheimer diseases (see Doulberis M et al, Helicobacter 2018). Please discuss.

Author Response

The authors thank Reviewer 3 for their constructive input. We added some key findings of the current literature to the abstract, of which we think it represents the state of the art and clarifies the focus of this review. Additionally, according to the reviewer's accurate suggestion, we changed the title of paragraph 2, so it outlines the following information more adequate. Since the body of knowledge described for Parkinson's disease was underrepresented in our review, we kindly thank Reviewer 3 for the suggestion to add further information. We extended this aspect, without going too much into detail, and hope it is sufficient for the reviewer. Furthermore, we tried to summarize the hypothetical pathomechanism by which H. pyloriis contributing to Alzheimer's disease, hopefully to your satisfaction. 

Reviewer 4 Report

Overall a well written and constructed review

The abstract was clear and covered the intentions of the study

The literature review was comprehensive and defined the area of inquiry

Likewise, the review focus and rationale for the review was well argued.

Each heading and review/discussion was logical and sequential and made the case well

I thought the range of issues addressed was impressive as was the argument made, the authors clearly have a strong understanding of the literature in this area.

I think this paper makes a contribution as it examines issues that are very current in microbiome research across a variety of topics.

This is one of the better papers I have reviewed and don’t have any areas of major concern or revisions and think it can be published. 

Author Response

We thank Reviewer 4 for reviewing our manuscript. 

Round  2

Reviewer 2 Report

The authors have adequately addressed my concerns, thus improving the quality of this work. I do not have further comments.

Author Response

We thank Reviewer 2 for reviewing our manuscript. 

Reviewer 3 Report

Page 5 line 237: Heliobacter --> Helicobacter

All other answers are satisfactory

Author Response

We thank Reviewer 3 for reviewing our manuscript and making us aware of this typo.